# Simulated range of mid-Holocene precipitation changes from extended lakes and wetlands over North Africa

Nora Farina Specht[1], Martin Claussen[1,2], and Thomas Kleinen[1]

[1]Max Planck Institute for Meteorology, Bundesstrasse 53, 20146 Hamburg, Germany

[2]Meteorological Institute, Centrum für Erdsystemforschung und Nachhaltigkeit (CEN), Universität Hamburg, Bundesstrasse 55, 20146 Hamburg, Germany

**Correspondence:** Nora Specht (nora-farina.specht@mpimet.mpg.de)

**Abstract.**

Enhanced summer insolation over North Africa induced a monsoon precipitation increase during the mid-Holocene, about 6000 years ago, and led to a widespread expansion of lakes and wetlands in the present-day Sahara. This expansion of lakes and wetlands is documented in paleoenvironmental sediment records, but the spatially sparse and often discontinuous sediment records provide only a fragmentary picture. Former simulation studies prescribed either a small lake and wetland extent from reconstructions or focused on documented mega-lakes only to investigate their effect on the mid-Holocene climate. In contrast to these studies, we investigate the possible range of mid-Holocene precipitation changes in response to a small lake extent and a potential maximum lake and wetland extent.

Our study shows that during the summer monsoon season, the African rain belt is shifted about 2 ° to 7 ° farther north in simulations with a maximum lake or wetland extent than in simulations with a small lake extent. This northward extent is caused by a stronger and prolonged monsoon rainfall season over the North Africa which is associated with an increased monsoon precipitation over the southern Sahara and an increased precipitation from tropical plumes over the northwestern Sahara. Replacing lakes with vegetated wetlands causes an enhanced precipitation increase which is likely due to the high surface roughness of the wetlands. A moisture budget analysis reveals that both, lakes and wetlands, not only cause a local precipitation increase by enhanced evaporation, but also by a stronger inland moisture transport and local moisture recycling to the south of Lake Chad and the West Saharan lakes. Analysis of the dynamic response show that lakes and wetlands cause a circulation response inverted to the one associated with the Saharan heat low. Depending on the latitudinal position of the lakes and wetlands, they predominantly cause a northward shift or a decay of the African Easterly Jet. These results indicate that the latitudinal position of the lakes and wetlands strongly affects the northward extension of the African summer monsoon.

## 1 Introduction

Paleoenvironmental sediment records reveal that North African lakes and wetlands spatially expanded during the mid-Holocene, as a result of increased summer monsoon precipitation (Holmes and Hoelzmann, 2017; Lézine et al., 2011). This precipitation increase was initiated by changes in the orbital forcing Kutzbach (1981), but reinforced by surface changes such as the expansion of vegetation (Kutzbach et al., 1996; Claussen and Gayler, 1997), the formation of soil (Levis et al., 2004; Vamborg

et al., 2011) and the extent of lakes and wetlands (Coe and Bonan, 1997; Broström et al., 1998; Carrington et al., 2001; Krinner et al., 2012; Chandan and Peltier, 2020). Even though the extent of lakes and wetlands during the mid-Holocene is visible in sediment records, these records are spatially sparse and most of them are temporally discontinuous (Holmes and Hoelzmann, 2017; Lézine et al., 2011). Additionally, reconstructions widely differ regarding the existence of mega-lakes (Quade et al., 2018). Given this spatially and temporally limited information from reconstructions, investigating the effect of lakes and

wetlands on the mid-Holocene climate becomes a scientific challenge.

Previous simulation studies used different approaches to prescribe mid-Holocene lakes and wetlands to investigate their effect on the North African climate. The majority of these studies prescribed a small lake and wetland extent using the reconstruction map by Hoelzmann et al. (1998) (Broström et al., 1998; Carrington et al., 2001; Krinner et al., 2012). Results from these investigations show that a small lake and wetland extent only causes a marginal northward shift of the North African

rain belt (Coe and Bonan, 1997; Broström et al., 1998; Carrington et al., 2001). A larger shift is caused, though, when this initial precipitation response is reinforced by vegetation feedback (Krinner et al., 2012). A more recent simulation study by Chandan and Peltier (2020) that considers several documented mega-lakes (e.g. Lake Ahnet, Chotts, Fezzan, Dafur and Chad) indicates that these large lakes only have a little impact on the northward penetration of the North African rain belt, but induce a precipitation increase over the Sahel region. These prior simulation studies all follow the approach of prescribing lakes and

wetlands that are documented by reconstructions.

Comparison between dust emission simulations and marine sediment cores indicate that mid-Holocene lakes and wetlands might have expanded much stronger over the Western Sahara (Tegen et al., 2002; Egerer et al., 2018) than prescribed in any mid-Holocene simulations (Hoelzmann et al., 1998; Chandan and Peltier, 2020). Moreover, little research has been done regarding the role of vegetated wetlands during the mid-Holocene. Vegetated wetlands have been prescribed in the vicinity

of mega-lake Chad (Carrington et al., 2001; Hoelzmann et al., 1998), yet the existence of vegetated wetlands in the vicinity of other documented mega-lakes is also conceivable. In addition, vegetated wetlands may have formed as a result of seasonal flooding, as seen in the Okavango Delta in South Africa.

Thus, the limited availability of mid-Holocene sediment records entail interpretative uncertainties in the effect of lakes and wetlands on the mid-Holocene climate. Therefore, we investigate the possible range of mid-Holocene precipitation changes

by exploring the effect of a small (Hoelzmann et al., 1998) and a potential maximum lake and wetland extent (Tegen et al., 2002), rather than following the approach of prescribing lakes and wetlands documented in reconstructions. This includes larger lakes and wetlands over the western Sahara, which might strongly affect the Saharan heat low, a shallow low pressure system, where near-surface monsoon southwesterlies and dry desert northeasterlies converge (Nicholson, 2009). The Saharan heat low controls the northward penetration of the monsoon winds (?) and, thus, the West African monsoon precipitation itself.

## 2  Methods

In order to investigate the potential range of mid-Holocene precipitation changes induced by extended lakes and wetlands, we conduct four mid-Holocene sensitivity experiments. The sensitivity experiments are performed using the atmosphere model

ICON-A (Giorgetta et al., 2018) and the land model JSBACH4 (Schneck et al., 2021; Reick et al., 2021) at ∼160 km horizontal resolution and 47 vertical hybrid sigma level. This coupled atmosphere-land model is forced with climatological 6 kyr BP orbital parameters (Berger, 1978) and greenhouse gas concentrations (Brovkin et al., 2019). The 6 kyr BP climatological vegetation distribution is prescribed based on a transient mid-Holocene MPI-ESM simulation (Dallmeyer et al., 2021).

The ICON-ESM simulates regional sea-surface temperatures (SST) biases of up to 5 K (Jungclaus et al., 2021), which causes a latitudinal displacement of the North African summer monsoon (Zhao et al., 2007). In order to simulate the latitudinal position of the North African summer monsoon in better agreement with proxy data, we use observation-based AMIP2 data to prescribe the SST and sea-ice concentration (SIC) (Taylor et al., 2000; Kanamitsu et al., 2002). The SST and SIC boundary conditions are derived as the following: monthly climatological differences between a 6 kyr BP (Jungclaus et al., 2019; Brierley et al., 2020) and historical (1980-2014) (Wieners et al., 2019) MPI-ESM PMIP4-CMIP6 simulation are superimposed onto a AMIP2 climatology (1980-2014). To take into account the seasonal SST and SIC variability, transient monthly SST and SIC anomalies from the 6 kyr BP simulation are added to the AMIP2-like mid-Holocene climatology. Since Sahel rainfall responds non-linearly to SST changes (Neupane and Cook, 2013), particularly over the tropical Atlantic (e.g. Rodríguez-Fonseca et al. (2015)), adding transient monthly anomalies to the climatological monthly mean SST, affects the overall mean state of the North African monsoon system.

Using this setup, the individual sensitivity experiments are run for 35 years, in which only the last 30 years are used as evaluation period. The individual sensitivity experiments differ only in their lake and wetland extent over North Africa: (1) present lakes, (2) small lake extent, (3) maximum lake extent, (4) maximum wetland extent. In the control simulation, the present lake map for standard ICON AMIP2 simulations is used (Fig. 1a). The small lake extent is given by the reconstruction map from Hoelzmann et al. (1998) that is based on paleo-environmental records (Fig. 1b). The maximum lake extent is represented by the potential maximum lake extent simulated by Tegen et al. (2002) using the hydrological model HYDRA (Fig. 1c). The same potential maximum extent map is also used to prescribe the maximum extent of wetlands (Fig. 1c).

The lakes and wetlands are treated as fractional grid cell types in the land model JSBACH4. The lakes are represented by a simple constant-depth mixed-layer (10 m) approach (Roeckner et al., 2003). The surface temperature of the lakes is uniquely dependent on the net surface heat flux, and the lake albedo is set to a constant value of 0.07 (Roeckner et al., 2003). The wetland surface cover type was implemented into JSBACH4 for the purpose of this study and is defined as moisture saturated soil with an equal ratio of C3 and C4 grasses growing on it. This representation of wetlands is similar to the one used by Carrington et al. (2001). In contrast to lakes, wetlands have a higher surface roughness and a dynamic background albedo that depends on the vegetation's net primary productivity.

As indicated in the former section, the changes of the Saharan desert ground due to litter production from the vegetation is taken into account. Previous simulation studies showed that these changes in the background albedo (surface albedo without vegetation) substantially influence the mid-Holocene precipitation over the Sahel/Sahara region (Vamborg et al., 2011). To

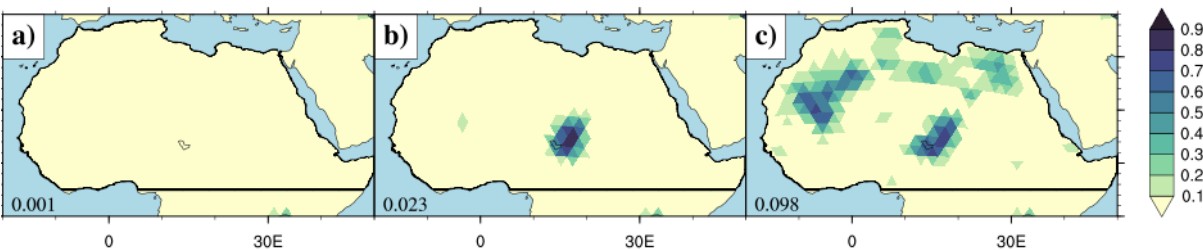

**Figure 1.** (a) standard AMIP2 present-day lake fraction from the JSBACH4-ICON model, (b) mid-Holocene small lake fraction derived from paleoecological reconstructions (Hoelzmann et al., 1998), and (c) mid-Holocene maximum lake fraction derived using the hydrological routing algorithm (HYDRA) (Tegen et al., 2002). The present-day lake fraction of Lake Chad is less than 10% per grid box, which falls into the lowest subsection of the plotting scale. The labels in the lower left corner of the plots indicate the lake fraction averaged over the North African region (5 °N (black line)−38°N and 20 °W−50 °W).

prescribe these background albedo changes, we use a parameterization scheme that is based on the concept by Egerer et al. (2018):

$$\alpha_{background} = \alpha_{mineral} - (\alpha_{mineral} - \alpha_{soil}) \times MIN\left(\sum_{PFT}^{i=1} f_i \frac{\overline{NPP_i}}{NPP_{soil,i}}, 1.0\right) \qquad (1)$$

The scheme by Egerer et al. (2018) prescribes the albedo changes of a mineral desert ground ($\alpha_{mineral}$) towards a organic soil ground $\alpha_{soil}$ due to the production of dead biomass from the vegetation in the mineral ground. The background albedo changes depend linearly on the 5-year mean net primary productivity $\overline{NPP}$ of the vegetation relative to the fixed annual NPP needed to completely cover the ground with one layer of dead organic material $NPP_{soil}$. By considering this linear relation Eq. (1), the scheme by Egerer et al. (2018) neglects non-linear processes such as the overlapping of dead leaves and litter with increasing coverage of the ground with organic material. Therefore, we here use the following exponential function to describe the background albedo changes during the mid-Holocene:

$$\alpha_{background} = \alpha_{mineral} - (\alpha_{mineral} - \alpha_{soil}) \times e^{-\sum_{PFT}^{i=1} f_i \frac{\overline{NPP_i}}{NPP_{soil,i}}} \qquad (2)$$

The soil albedo $\alpha_{soil}$ is set to a mean value of 0.13 for the visible range and a mean value of 0.22 for the near-infrared range as suggested by Egerer et al. (2018). $\alpha_{mineral}$ is derived by inverting Eq. (2) and then calculating $\alpha_{mineral}$ for each grid cell using $\overline{NPP}$ from a standard AMIP ICON simulation and $\alpha_{background}$ obtained from MODIS observation data (Otto et al., 2011). Since the background albedo scheme prescribed the transition from a mineral desert ground to a savanna landscape,

this scheme is applied only in the North African Sahel-Sahara region (10 °N-35 °N; 20 °W-35 °W). Results reveal that the exponential parameterization scheme is able to capture the relation between background albedo and NPP (not shown).

In this study, the mid-Holocene climate response to the extended lakes and wetlands is calculated by the 30-year mean difference between the control simulation (present-day lakes) and the sensitivity simulations (extended lakes and wetlands):

$$\delta(\cdot) = (\cdot)_{sensitivity} - (\cdot)_{control} \tag{3}$$

Under this stationary state, the moisture budget changes can be divided into:

$$\delta P = \delta E + \delta(P - E) \tag{4}$$

where $\delta P$ is the precipitation response, which consists of a local evaporation response $\delta E$ and a moisture convergence response $\delta(P - E)$ that indicates local drying or wetting. Following Seager et al. (2010), the moisture convergence response can be separated into a dynamic $\delta DY$, a thermodynamic $\delta TH$, a transient eddy $\delta TE$, a non-linear $\delta NL$ and a surface term $\delta S$. In this study, the $\delta S$ is relatively small, so that the moisture convergence response can be expressed as:

$$\delta(P - E) \approx \delta DY + \delta TH + \delta TE + \delta NL \tag{5}$$

$$\delta DY = -\frac{1}{g\rho_w} \int_0^{p_s} \nabla \cdot ([\delta \overline{u}]\overline{q}_{control}) dp \tag{6}$$

$$\delta TH = -\frac{1}{g\rho_w} \int_0^{p_s} \nabla \cdot (\overline{u}_{control}[\delta \overline{q}]) dp \tag{7}$$

$$\delta TE = -\frac{1}{g\rho_w} \int_0^{p_s} \nabla \cdot \delta(\overline{u'q'}) dp \tag{8}$$

$$\delta NL = -\frac{1}{g\rho_w} \int_0^{p_s} \nabla \cdot (\delta \overline{u} \delta \overline{q}) dp \tag{9}$$

where an overbar indicates a monthly mean and, a primed variable, deviation from the monthly mean. $\delta DY$ (Eq. 6) represent changes in the moisture convergence due to changes in the mean circulation $\delta \overline{u}$ and $\delta TH$ (Eq. 7) indicates changes in the

moisture convergence due to changes in the mean specific humidity $\delta\overline{q}$. The $\delta TE$ (Eq. 8) and $\delta NL$ (Eq. 9) account for moisture convergence changes caused by coherent changes in the wind and specific humidity. The terms of the moisture budget equation provide information about the mechanisms that affect the simulated precipitation response induced by the prescribed extended lakes and wetlands.

## 3  Spatial and temporal precipitation response

The maximum lake extent shifts the North African rain belt about 2 ° to 3 ° farther north into the Sahel region (11 °N to 17 °N) than the small lake extent (Fig. 2). The line between savanna and desert landscape, marked by the Sahel rainfall threshold (200 mm/year), is shifted about 7 ° farther north in the simulation with the maximum lake extent than in the simulation with the small lake extent (Fig. 2). Comparison of the maximum lake and wetland experiments shows that replacing lakes with vegetated wetlands leads to an overall increase of precipitation over North Africa (Fig. 2). The relative precipitation difference
is particularly strong over the northern Sahara (22 °N-30 °N).

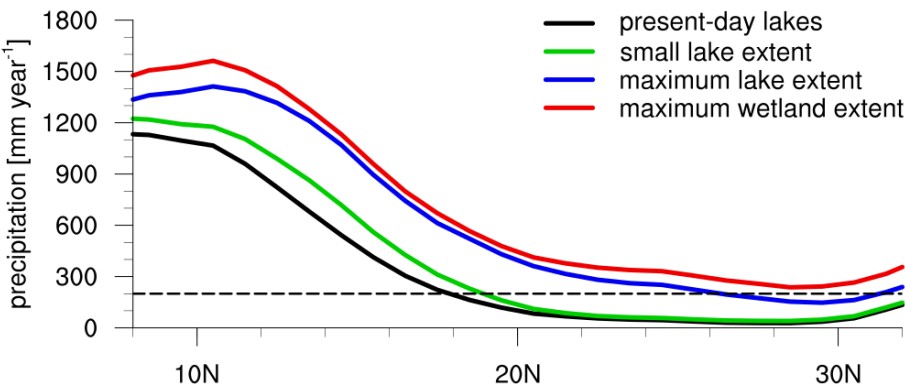

**Figure 2.** Mid-Holocene 30-year mean annual precipitation zonally averaged over Africa (10 °E – 30 °W). The black dashed line marks the 200 mm/year threshold indicating the border between savanna and desert landscape.

The annual course of the zonally averaged daily precipitation shows that the expansion of lakes and wetlands prolongs the rainfall season over North Africa (Fig. 3 a-d). Hagos and Cook (2007) define the onset and offset of the rain season over North Africa as a jump of the precipitation maximum from 5 °N (near the Gulf of Guinea) to 10 °N. Accordingly, the duration of the North African rainfall season is marked by the period in which the core of maximum rainfall lies above 10 °N (black solid
line in Fig. 3 a-d). Results show that, by this definition, the rainfall season is almost twice as long in the maximum lake and wetland simulations than in the control simulation. Prescribing a small lake extent in the mid-Holocene simulation, however, only causes marginally longer duration of the North African rainfall season.

In all sensitivity experiments, the earlier onset and later offset of the monsoon season is associated with an enhanced monsoon precipitation over the rain belt region between the Tropical Easterly Jet (TEJ) and the African Easterly Jet (AEJ) (Fig. 3). In

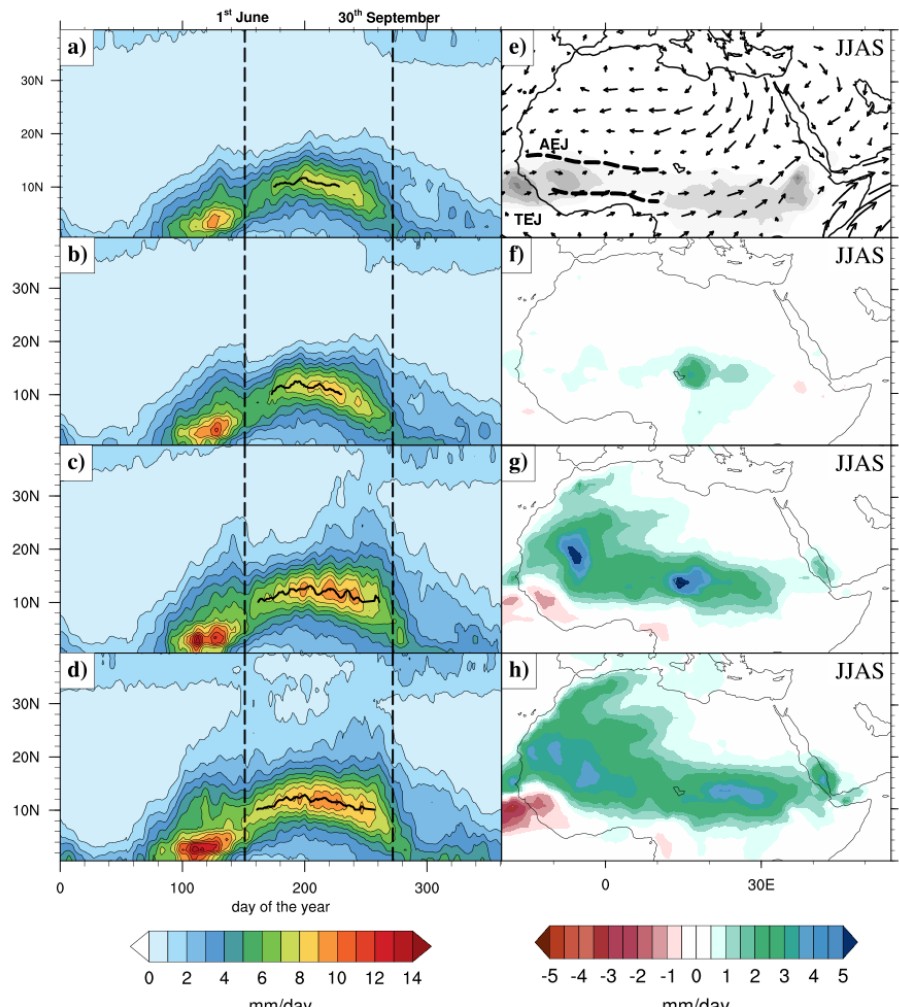

**Figure 3.** (a-d) Mid-Holocene 30-year mean daily precipitation zonally averaged over Africa (10 °E – 30 °W) and smoothed with a 5-day running mean. The black lines represent the time period in which the core of maximum rainfall lies above 10 °N and, thus, indicates the duration of the rain season over the Sahel. (e) Schematics of 30-year mean JJAS African summer monsoon features from the present-day lake simulation: (arrows) the near-surface wind (at 850 hPa), the mid-troposphere African Easterly Jet (AEJ) (600 hPa), the Tropical Easterly Jet (TEJ) (150 hPa) and (grey shades) the rain belt region between both jets. (f-g) Mid-Holocene 30-year mean JJAS precipitation response to the (f) small lake extent, (g) maximum lake extent and (h) maximum wetland extent in comparison to the present-day lake simulation.

the maximum lake and wetland experiments, the prolonged rainfall season is also associated with an increased rainfall over the northwestern Sahara (20 °N-35°N) during September (Fig. 3 c-d and g-h). In the maximum wetland experiment, this enhanced rainfall over the northwestern Sahara also occurs at the rain season onset in June (Fig. 3 h).

Beside seasonal changes, we also analyze the spatial precipitation response to the prescribed extended lakes and wetlands for the summer months (JJAS). Results show that the small and the maximum lake extent, both, cause a local prescription increase over the southern Saharan (Fig. 3 f-g). The maximum lake extent, additionally, causes an enhanced monsoon precipitation increase over the rain belt region (between the AEJ and TEJ), as well as increased precipitation over the center of tropical-extratropical interaction in the northwestern Sahara (Fig. 3 e and g). Comparison of the maximum lake and maximum wetland experiments shows that replacing the lakes with vegetated wetland leads to an overall increase of the summer monsoon precipitation over North Africa. The vegetated wetlands cause a spatially broader precipitation increase, compared to the local precipitation peak induced by the lakes (Fig. 3 g-f).

## 4   Moisture convergence response

In order to better understand the influences of the lakes and wetlands on the mid-Holocene precipitation, changes in the moisture budget terms are analyzed. Changes in evaporation and moisture convergence, both, substantially contribute to the summer precipitation increase over North Africa (Fig. 4). A local evaporation increase over the prescribed lakes and wetlands corresponds to an enhanced precipitation over the northern Sahara and Lake Chad (Fig. 4 a-c). The moisture convergence response shows a dipole pattern with a drying response over the northern Sahara and north of Lake Chad and a wetting response over the southern Sahara, the Sahel region and south of Lake Chad (Fig. 4 d-f). Results show that evaporation changes are the major cause for the precipitation increase over the northern Sahara and over Lake Chad, whereas enhanced moisture convergence primarily contributes to the rainfall increase over the southern Sahara and Sahel region.

The moisture divergence and, thus, drying response over the northern Sahara and north of Lake Chad counteract the effect of increased evaporation in this region (Fig. 4). In the maximum lake and wetland experiments, the drying response over the northeastern Sahara completely compensates the local effect of increased evaporation (Fig. 4 b-c and e-f). Accordingly, the summer precipitation is unchanged over the northeast Saharan lakes and wetlands in the maximum lake and wetland experiments (Fig. 3 g-h).

Results also show that the latitudinal position of the wet-south-dry-north dipole pattern differs between the small lake extent and the maximum lake and wetland extent experiments. The boundaries between wetting and drying response is related to the latitudinal position of the prescribed lakes and wetlands over the western Sahara and the Chad region. In the maximum lake and wetland experiments, the line between wet and dry response reaches up to 20 °N over the western Sahara, whereas this line is restricted to 15 °N by Lake Chad in the small lake experiment.

In order to investigate the causes for this dipole-like moisture convergence response $\delta(P-E)$ over North Africa, $\delta(P-E)$ is split into a dynamic $\delta DY$, thermodynamic $\delta TH$, transient eddy $\delta TE$ and non-linear term $\delta NL$ following Seager et al. (2010). Results indicate that the dipole-like response is primarily caused by dynamic and thermodynamic changes that affect the moisture convergence (Fig. 4 d-i). Therefore, we only analyze the $\delta DY$ and $\delta TH$ in detail.

The thermodynamic response $\delta TH$ reflects the near-surface convergence-divergence pattern associated with the mean summer monsoon circulation (Fig. 5 a-c). This response is associated with a general increase in the near-surface mean specific

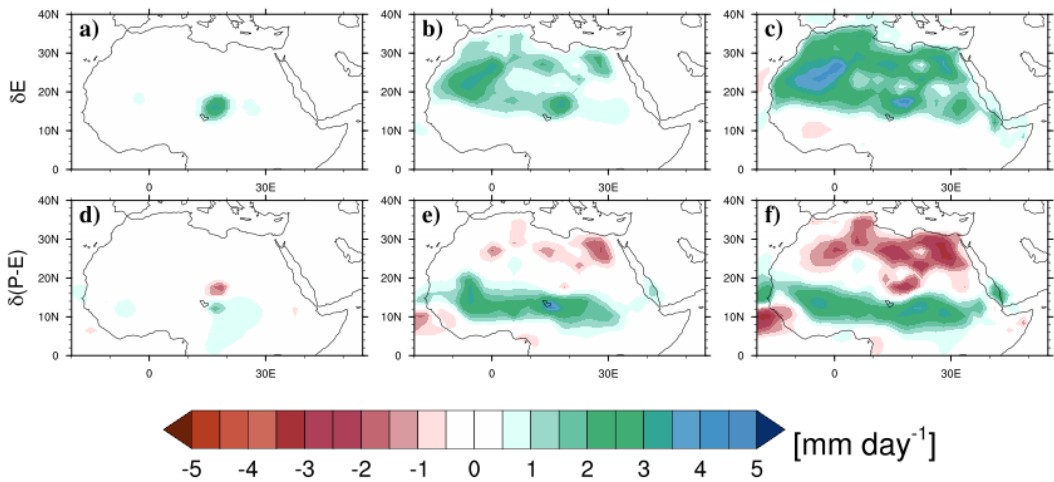

**Figure 4.** Mid-Holocene 30-year mean summer (JJAS) (a-c) evaporation and (d-f) moisture convergence response to a (left) small lake extent, (middle) maximum lake extent and (right) maximum wetland extent. The sum of the evaporation and moisture convergence response is equal to the precipitation response shown in Fig.3 f-h.

humidity caused by the prescribed lakes and wetlands. Increased moisture convergence occurs to the south and north of the AEJ, particularly near the African West coast (Fig. 5 a-c). This region is associated with the southern and northern track of the African Easterly Waves (e.g Nicholson (2009)). Increased moisture convergence also occurs near the northwest Saharan coast (Fig. 5 b-c), where the cyclonic flow from the Saharan heat low and the anticyclonic flow from the extra-tropical Azores high converge (e.g Nicholson (2009)). Moisture divergence and, thus, a drying response is visible over the northeastern Sahara (Fig. 5 a-c), where hot and dry northeasterlies, known as Harmattan winds, likely transport moisture away from the lakes and wetlands (Fig. 3 e). Comparison of the individual sensitivity experiments shows that $\delta TH$ is relatively weak in the small lake extent experiment compared to the maximum lake extent experiment. The maximum wetland extent causes generally larger changes of $\delta TH$ than the maximum lake extent. The changes in $\delta TH$ relate to the strength of the evaporation increase and, thus, the increase in the specific humidity caused by the prescribed lake and wetlands extent in the individual sensitivity experiments.

The dynamic response $\delta DY$ reveals that the prescribed lakes and wetlands enhance the moisture convergence in the southern West Saharan and the southern Chad region (Fig. 5 a-c). This increased moisture convergence is associated with stronger near-surface monsoon westerlies that enhance the inland moisture transport from the Atlantic ocean to the North African continent (Fig. 5 a-c). A weakening of the AEJ (Fig. 6 b-c) additionally contributes to this moisture convergence increase by decreasing the moisture export from the North African continent to the Atlantic ocean. A comparison between the small and large lake experiments shows that the maximum extent of the lake induces a stronger monsoon westerly wind acceleration that reaches

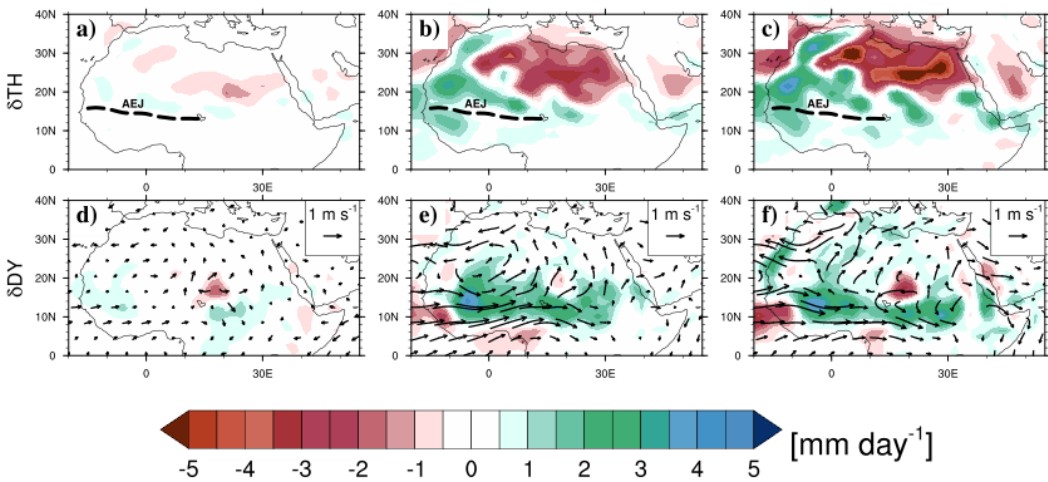

**Figure 5.** Mid-Holocene 30-year mean summer (JJAS) (a-c) thermodynamic $\delta TH$ and (d-f) dynamic moisture convergence response $\delta DY$ to a (left) small lake extent, (middle) maximum lake extent and (right) maximum wetland extent. Changes in the moisture convergence (precipitation response minus evaporation response) are primarily caused by changes in the dynamic component $\delta DY$ and thermodynamic component $\delta TH$.

farther to the north (Fig. 5 a-b). Accordingly, the moisture convergence increases stronger and reaches higher up north in the maximum lake experiment (Fig. 5 a-b). Since these differences are particularly strong over the western Sahara, the west Saharan lakes are likely the primary cause for the stronger inland moisture transport seen in the maximum lake extent experiment. Besides, the size of Lake Chad is nearly similar in the small and maximum lake extent experiments (Fig. 1 b-c). The dynamic response to the maximum lake and wetland extent shows similar changes in the mean monsoon circulation over the southern Sahara, which results in a similar $\delta DY$ pattern (Fig. 4 e-f).

The non-linear response $\delta NL$, causes a reversed, but less pronounced response pattern compared to the thermodynamic response (Fig. A1 a-c) and is, therefore, overcompensated. The transient eddy response $\delta TE$ causes a moisture convergence decrease over the West Saharan lakes (around 25 °N) in the maximum lake and wetland extent experiments (Fig. A1 e-f), but shows only marginal changes in other regions (Fig. A1 d-f). Thus, the total moisture convergence changes primarily emerge from the dynamic and thermodynamic term of the moisture budget equation.

## 5   Atmospheric circulation response

In the sensitivity experiments, the largest precipitation increase occurs in the Chad region and western Sahara. Therefore, we analyze the circulation response to Lake Chad in the small extent experiment and the circulation response to the West Saharan lakes and wetlands in the maximum extent experiments.

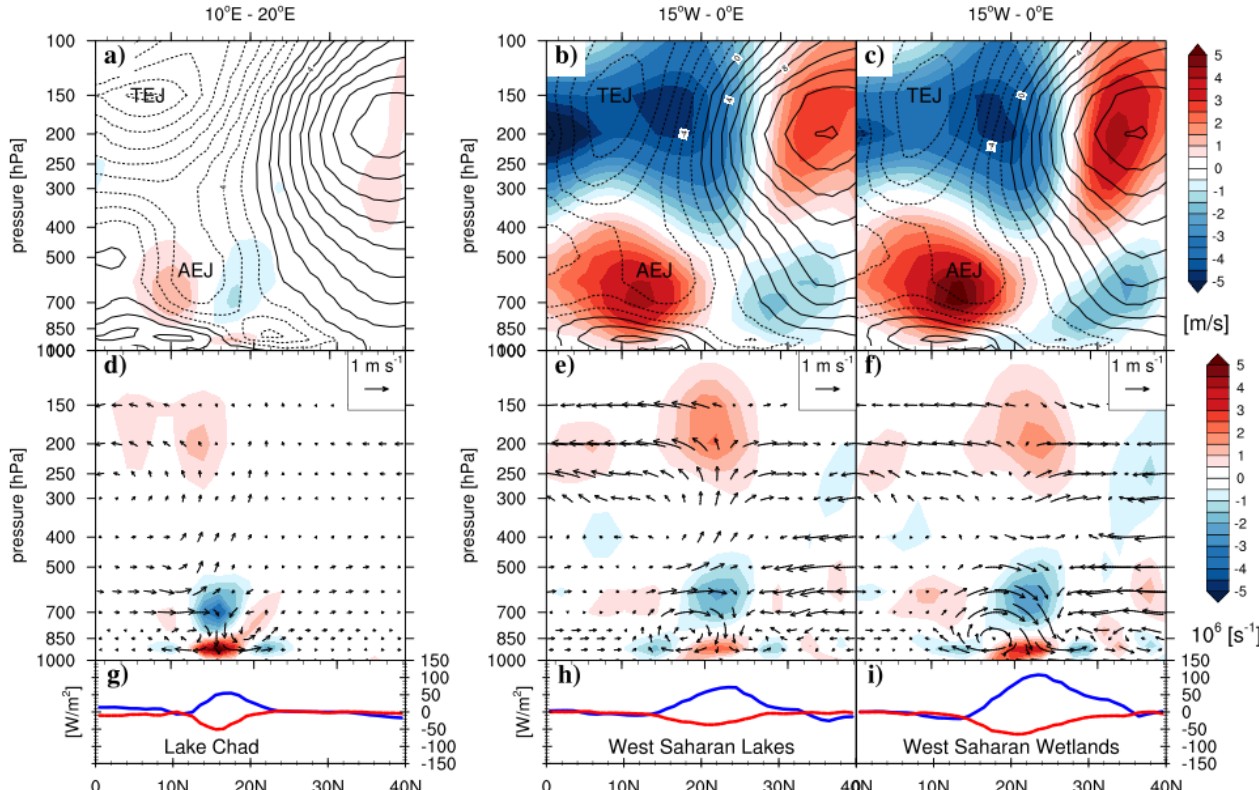

**Figure 6.** Mid-Holocene 30-year mean summer (JJAS) circulation response: (a-c) zonal wind response (colored shading), (d-f) divergence response (colored shading) and vertical wind response (black arrows) and (g-i) latent (blue) and sensible (red) heat flux response to the (left) small lake extent (zonally averaged over the Chad region: 10 °E – 20 °E), (middle) the maximum lake extent and (right) the maximum wetland extent (zonally averaged over the Western Sahara: 15 °W – 0 °E); (a-c) black contours show the mid-Holocene 30-year mean summer (JJAS) zonal wind climatology zonally averaged over the Chad and West Saharan region.

Lake Chad and the West Saharan lakes and wetlands generally cause a northwards shift of the North African summer monsoon which is associated with enhanced monsoon westerlies and a northward shifted and weakened AEJ (Fig. 6 a-c).
Evaporative cooling above the lakes and wetlands induces a circulation response which is inverted to the one associated with the Saharan heat low (Fig. 6 d-f). A decreased sensible heat flux and increased latent heat flux from the surface to the atmosphere causes descending motions over the cold the lakes and wetlands and ascending motions in their warmer vicinity. The resulting overturning circulation response is associated with a near surface divergence response and a convergence response aloft, at the mid-troposphere. The convergence increase and, thus, low pressure response at the mid-troposphere induces, by Coriolis force,
a westerly wind response to the south and an easterly wind response to the north of the lakes and wetlands at about 700 hPa (Fig. 6 a-c).

The location of this dipole-like zonal wind response depends on the latitudinal position of the lakes and wetlands (Fig. 6 a-c). The existence of Lake Chad leads to a dipole-like zonal wind response around 14 °N, which corresponds to a northward shift of the AEJ. The West Saharan lakes and wetlands induce a dipole-like zonal wind response at around 23 °N, which corresponds primarily to a strong weakening of the AEJ over the western Sahara. Accordingly, lakes located near the AEJ induce a northward shift of the AEJ, while lakes and wetlands located north of the AEJ primarily weaken the AEJ.

Beside lower level circulation changes, lakes and wetlands enhance the convergence of moist air at the mid-troposphere which favors moist convection in the mid- to upper troposphere above the lakes and wetlands (Fig. 6 d-f). The associated upper level divergence and, thus, high pressure response lead to a easterly wind acceleration to the south and westerly wind acceleration to the north of the West Saharan lakes and wetlands (Fig. 6 b-c). Accordingly, the TEJ is strengthened in the maximum lakes and wetlands experiments. Lake Chad, in contrast, only causes marginal changes in the upper level mean zonal wind.

Wetlands cause an overturning circulation response that extends higher into the troposphere than the overturning circulation response induced by lakes (Fig. 6 e-f). Accordingly, wetlands cause a higher convection increase than lakes in their vicinity. In contrast, lakes induce a stronger convection increase than wetlands above their water surface in the mid- to upper troposphere. The strong convection increase above the lakes and the strong convection increase in the vicinity of the wetlands likely causes the prescription peak over the prescribed lakes and a rather broad rainfall increase over and around the wetlands (Fig. 3 g).

## 6   Discussion and Conclusions

Our study corroborates the results of previous studies, which show that lakes in the Sahara likely enhanced the West African summer monsoon during the mid-Holocene (Krinner et al., 2012). The simulated rain belt northward shift of about  1 ° in the small lake extent experiment is relatively large compared to the northward shift simulated by most studies that prescribe a similar small lake extent (Coe and Bonan, 1997; Broström et al., 1998) or an even larger lake extent (Chandan and Peltier, 2020). Only the study by Krinner et al. (2012) shows an northward shift of 1.5 ° in response to this small lake extent. In contrast to other studies, Krinner et al. (2012) use a dynamic vegetation that presumably reinforces the precipitation response to the prescribed lakes. Since the vegetation is prescribed in our experiments, the precipitation response to the extended lakes and wetlands in our study can likely be attributed to background and evaporation feedback.

Our results show that the maximum lake extent shifts the isohyets of the mid-Holocene rain belt about 2 ° to 7 ° farther north than the small lake extent. This difference in the northward shift is mainly caused by the additional lakes over the western Sahara, while the prescribed Lake Chad is nearly of similar size in both experiments. The West Saharan lakes strongly affect the monsoon circulation because they lie in a region where the AEJ is strongest and where the Saharan heat low is located. Thus, the location of the prescribed lakes and wetlands likely plays a major role for the northward extent of the African summer monsoon. These findings disagree with the results of Chandan and Peltier (2020) who conducted two mid-Holocene simulation: one with reconstructed lakes and one with a homogeneous lake extent over North African (plus Lake Chad). We suppose that

the difference between our simulations results can be attributed to the fact that in their simulation, large lakes over the Western Sahara are missing.

Our study also shows that that vegetated wetlands cause a stronger precipitation increase than lakes of the same size. This strong precipitation response to vegetated wetlands is likely caused by a high surface roughness which is associated with an enhanced evaporation and surface cooling over the wetlands. Due to this strong surface cooling and, thus, local subsidence, wetlands cause a high precipitation increase in their vicinity. In contrast, lakes show a local precipitation peak above their water surface.

Analysis of the annual course of precipitation shows that the maximum lake and wetland extent over North Africa causes a prolonged rainfall season. This prolonged rainfall season is associated with an earlier onset and later offset of the monsoon season over North Africa. While an enhanced summer monsoon precipitation in the rain belt region occurs in all sensitivity experiments, the maximum lake and wetland extent additionally causes an increased precipitation over the northwestern Sahara during spring and autumn. Precipitation in the northwestern Sahara during fall is presumably associated with the occurrence of rain-bringing tropical plumes. These elongated cloud bands result from extratropical-tropical interaction between an extra-tropical Atlantic trough and the Saharan heat low. A model study by Skinner and Poulsen (2016) showed that increased moisture availability throughout North Africa during the mid-Holocene enhanced the formation of rainfall-producing tropical plumes over the northwestern Sahara which prolongs the rainfall season. Our results indicate that the west Saharan lakes and wetlands provide moisture for the formation of tropical plumes by increasing the local evaporation and by shifting the North African rain belt northward, which likely prolongs rainfall season.

Analysis of the moisture budget shows that the simulated precipitation increase in the sensitivity experiments is primarily caused by a local evaporation increase of the lakes and wetlands over the northwestern Sahara and by a enhanced moisture convergence over the southern Sahara and Sahel region. The increased precipitation over the southern Sahara is caused by enhanced inland moisture transports and local moisture recycling to the south of the west Saharan lakes and lake Chad. In contrast, moisture advection by the Harmattan wind transport moisture away from the lakes and wetlands over the northeastern Sahara. This local drying response over the northeastern Sahara indicates why lakes and wetlands do not enhance the local precipitation in this region. The wet-south dry-north moisture convergence response indicate that lakes and wetlands received excess water primarily from the southern part of their catchment during the summer months. A simulation study by Skinner and Poulsen (2016) and Dallmeyer et al. (2020) showed that the enhanced occurrence of rain-bringing tropical plumes during the mid-Holocene fall season substantially increased the annual precipitation over the western Sahara. Thus, West Saharan lakes and wetlands also might have received excess water during the fall season.

The dynamic response in sensitivity experiments shows that lakes and wetlands cause a circulation response inverse to the one associate with the Saharan heat low. Evaporative cooling causes descending motions above the lakes and wetlands, whereas ascending motions occur in their warmer vicinity. This overturning circulation response causes a convergence increase in the a mid-troposphere and, by Coriolis force, an westerly wind response to the south and an easterly wind response to the north of the lakes and wetlands. This dipole-like zonal wind response induces either a northward shift of the AEJ or a weakening of the AEJ depending on the latitudinal position of the lakes and wetlands. E.g. Lake Chad causes a northward shift of the

AEJ, whereas the higher up north located West Saharan lakes nearly cause a strong decay of the AEJ. Along with this shift or weakening of the AEJ the near-surface monsoon westerlies extent farther to the north. These circulation responses indicate that at the latitudinal position of the lakes and wetlands strongly affects the monsoon circulation.

Finally, our study shows that the border between savanna and Sahara, the 200 mm/year isohyet, is shifted by about 7 ° farther north in the maximum extent experiment than in the small extent experiment. This implies that lakes might had a substantial influence on the vegetation expansion during the mid-Holocene. In this study, we prescribed the vegetation, lakes and wetlands and thus, neglect any dynamic feedback. Yet, this dynamic interaction might play a key role in understanding the termination of the African Humid Period. Mid-Holocene reconstructions (Shanahan et al., 2015; Lézine et al., 2011) and transient mid-Holocene simulations (Dallmeyer et al., 2020) reveal that the African Humid Period ended earlier in the East Sahara than in the West Sahara. This immediately raises questions about the different time-scales of vegetation dynamics and lake dynamics and how they interactively affect the termination of the African Humid Period. Hence in a follow-up study we will explore the effect of dynamic lakes on the African Humid Period and how dynamic lakes and a dynamic vegetation interact with each other in this context.

*Data availability.* The paleo-environmental lake reconstruction map by Hoelzmann et al. (1998) - used to prescribe a small lake extent - is accessible at the National Center for Environmental Information of the National Oceanic and Atmospheric Administration (https://www1.ncdc.noaa.gov/pub/data/paleo/pollen/africa6k). The potential maximum lake area - used to prescribe a maximum lake and wetland extent - originates from the simulation study by Tegen et al. (2002). The potential maximum lake area data are accessible at T63 resolution at the German Climate Computing Center (https://cera-www.dkrz.de/WDCC/ui/cerasearch/entry?acronym=DKRZ_LTA_060_ds00002) and were provided by Egerer et al. (2018). For the purpose of this study the T63 reference data were interpolated to a R2B4 ICON grid. A summary of the model version, the boundary conditions, scripts and output data used in this study is available at http://hdl.handle.net/21.11116/0000-0009-63B5-B.

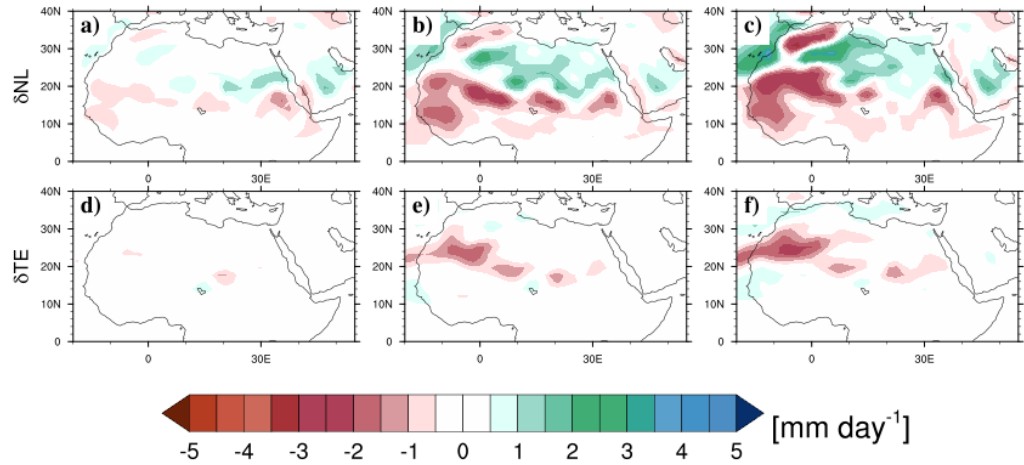

**Figure A1.** Mid-Holocene 30-year mean summer (JJAS) (a-c) non-linear and (d-f) transient eddy moisture convergence response to a (left) small lake extent, (middle) maximum lake extent and (right) maximum wetland extent.

*Author contributions.*  NFS and MC planned the study. NFS developed the model components, ran the simulations and analyzed the results. MC and TK contributed to the discussion of results and the manuscript.

*Competing interests.*  The co-author Martin Claussen is editor at Climate of the Past.

*Acknowledgements.*  This study contributes to the Cluster of Excellence EXC 2037 "Climate, Climate Change, and Society" (CLICCS),
funded by the Excellence Strategy of the German federal and state governments. T.K. is funded by the project PalMod of the German Federal Ministry of Education and Research (BMBF), Research for Sustainability initiative FONA (grant no. 01LP1921A). Nora Specht was financed by the International Max Planck Research School on Earth System Modeling (IMPRS-ESM), Hamburg. The model simulation were performed at the German Climate Computing Center (DKRZ). Many thanks go to Reiner Schnur (MPI-M) for his excellent technical support, to Thomas Raddatz (MPI-M) for his comprehensive expertise and internal review, to Philipp Hoelzmann (FU-Berlin) for the helpful
discussions and for providing the reconstruction data and to Ina Tegen for providing the model derived lake data. Special thanks also go to Roberta D'Agostino for conducting the 6 kyr BP MPI-ESM PMIP4-CMIP6 simulation and providing the SST and SIC data.

    The article processing charges for this open-access publication were covered by the Max Planck Society.

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
