# Peer review of "Simulated range of mid-Holocene precipitation changes from extended lakes and wetlands over North Africa"

_Climate of the Past, 2021_

## Author Comment (AC1)

**Reply to Referee comments on cp-2021-129, "Simulated range of mid-Holocene precipitation changes to extended lakes and wetlands over North Africa" by Specht et al.**

We thank the Referee #1 for the constructive and valuable comments to make this manuscript more understandable and to improve the results. In the following, we respond to each comment individually, in which the referee's comments are marked in black and our answer is marked in blue.

**Referee**: Thanks very much for assigning me to review this manuscript entitled *"Simulated range of mid-Holocene precipitation changes to extended lakes and wetlands over North Africa"*. The authors analyze the distinct responses of mid-Holocene precipitation to prescribe several surface boundaries conditions. Overall, this paper very interesting and the logical is clear. From my understanding, moderate revisions are needed before its publication on this journal.

**Answer**: Thank you for pointing out that the study is interesting and that the main story line is clear.

**Referee**:

Comments on this manuscript:

For the method part,

The authors should give more details of "moisture budget analysis and its thermodynamic and dynamic decompositions"used in this study, because not every reader is a good at dynamics.

**Answer**: We agree that a more detailed introduction of the moisture budget components would support the comprehensibility of the manuscript. Therefore, we will include the equation by Seager et al. (2010) into the methods part and explain these terms using simple examples, e.g. "the dynamic term prescribes changes in the moisture budget caused by a response in the mean circulation, e.g. changes in inland moisture transport by the monsoon westerlies."

**Referee**:

For the result part,

Please give one more Fig2b in the Fig2 which is related to"evaporation "responses, in order to support the authors statement "P4L108-110"
**Answer**: We will add an additional figure (map plot) that show the contribution of the local evaporation response and moisture budget response to the overall precipitation response.

**Referee**: Please use "local evaporation"instead of "direct evaporation"(P5L123)
**Answer**: Will be done.

**Referee**: Please take care the statement (P5L125). Indeed was neglected in Seager et al. 2010.
**Answer**: I am not sure what this comment refers to, but we will take care of explaining the moisture budget analysis and its meaning in the method part.

**Referee**: Please check the order of the Fig.3 and Fig.4, I am afraid the fig.3 should have appeared in some place prior to Fig.4.

**Answer***:* We will changes the order of figure 3 and 4.

**Referee**: If possible, please add the extent of change in each experiment so that it will Convery a very clear picture for the reader.

**Answer**: We will show the contribution of the local evaporation response and moisture budget response to the overall precipitation response in a separate figure (as mentioned above).

**Referee**: Typo error, ITZC L149

**Answer**: We will correct this typo.

**Referee**: Please add the horizontal circulation changes in each subplot of Figure 4 to support your statements P6L133-134

**Answer**: We will add the horizontal wind response at 850 hPa to figure 4 d-f.

---

## Author Comment (AC2)

**Reply to Referee comments on cp-2021-129, "Simulated range of mid-Holocene precipitation changes to extended lakes and wetlands over North Africa" by Specht et al.**

We want to thank Chris Brierley for his constructive comments and appreciate his suggestion on how to analyses the results more comprehensively. In the following, we respond to each comment, in which the referee's comments are marked in black and our answer is marked in blue.

**Referee**: This paper is obviously a good fit for publication in climate of the past. It describes some novel simulations looking at the role of lakes and wetlands over North Africa. There is clearly sufficient work represented by simulations to warrant a publication, and I commend the authors for having done these runs. The inclusion and analysis of the water budget is also a nice feature.

**Answer**: Thank you for the positive feedback on the simulations and the moisture budget analysis.

**Referee**: Unfortunately I don't feel this manuscript is worthy of publication in its current state, because of the open questions around seasonality. A quarter of the manuscript's Discussion section is devoted to speculation about the rainfall changes in autumn using other published work. This is further emphasized in the conclusions. If I've understood the methodology section correctly, then such information could be computed directly by the simulations. A revised manuscript should include some analysis of this data for autumn – preventing there being any need to speculate.

**Answer**: We agree that our discussion was too speculative. Therefore, we will include additional analysis on the seasonality of the precipitation response from the prescribed lakes and wetlands to provide a more comprehensive picture. The additional plot will show a substantial precipitation increase over Northwest Africa around the end of the rainy season.

**Referee**: Other minor comments.

The title should be reworded to say "from extended lakes", to better convey the direction of influence.

**Answer**: We will reword the title as suggested ("to" → "from")

**Referee**: I suggest changing lines L42-L44 from

"Moreover, little research has been done regarding the role of vegetated wetlands during the mid-Holocene (Carrington et al., 2001). Present investigations on the effect of vegetated wetlands prescribed a small extent in the western Sahara and in the vicinity of mega-lake Chad (Carrington et al., 2001; Hoelzmann et al., 1998). Apart from mega-lake Chad,"…

To

"Moreover, little research has been done regarding the role of vegetated wetlands during the mid-Holocene. Vegetated wetlands have been prescribed in the vicinity of mega-lake Chad (Carrington et al., 2001; Hoelzmann et al., 1998), yet"…

**Answer**: We rephrase this test part as suggested.

**Referee**: L47 "in mid-Holocene" should be "of mid-Holocene"

**Answer**: We will rephrase this text part as suggested.

**Referee**: L60 Whilst I appreciate you citing both of my papers here, I am unclear why our work on ENSO is relevant. Instead you might want to cite the doi number from the EGSF to give better credit to the MPI team who performed the mid-Holocene simulation.

**Answer**: Thank you for making this point. We have decided to take the SST and SIC from the PMIP simulation published in your paper. We will add citation of the DOI provided by the ESGF in the method part and data availability and we will acknowledge Roberta D'Agostino who did this simulation. Regarding vegetation, we choose to take the vegetation distribution, including the patterns of plant functional types, from a parallel 6k simulation, which is published and described in Dallmeyer et al. (2021).

**Referee**: L65 What is it necessary to the monthly SIC and SST anomalies from the MPI mid-Holocene simulation to your experimental design. I can see why you might do it, but you analyse nothing but the climatologies in this paper.

**Answer**: From our experience, we know that the simulated atmospheric mean state differs depending on whether only climatological SSTs and SICs were prescribed or climatological SSTs and SICs plus variability.

**Referee**: Fig 1. Add in the caption that the present-day lakes are also drawn. Can you make them more visible as well, please?

**Answer**: The Chad outline shown in figure 1 is drawn by default by the plotting tool NCL and does not show the present-day Chad lake prescribed in the control experiment. We will therefore add a third plot that shows the present-day lake distribution as prescribed in the experiments.

**Referee**: It is not clear how eq. 1 differs from that of Egerer et al. Please explain.

**Answer**: We will add the information that Egerer et al. 2018 uses the following linear approach: $MIN(NPP/NPP_{soil}, 1)$; whereas an inverse exponential approach is used in our study: $e^{(-NPP/NPP_{soil})}$. This was already stated in the manuscript, but obviously not explained very clearly.

**Referee**: L104: prescribed -> proposed

**Answer**: Okay.

**Referee**: L115: Please add "likely" before associated. You provide no evidence of this statement.

**Answer**: We agree on putting "likely" before statement L115

**Referee**: L116: It is unclear what evidence you are basing this claim on. Figure 2 shows only zonal mean values, and so cannot be used to support discussion of regional changes.

**Answer**: We will exclude this statement for the sake of clarity.

**Referee**: Figure three is barely mentioned and feels like a remnant of an earlier draft. It could be a useful figure.

**Answer**: One of our co-authors, whose daily business is not the AHP, requested this figure to improve clarity for the non-initiated readers.

**Referee**: L155. The "Sahara heat low" is an undefined term. I find myself thinking of low heats and cold places.

**Answer**: The Saharan heat low is a standard meteorological term (e.g. Claussen et al. 2017; Nicholson, 2009) .

**Referee**: Please label Fig 4 as mm/day.

**Answer**: Okay.

**Referee**: L165: figure 5 doesn't really show a northward shift of the African easterly jet. Rather it shows a death of the southern portion of it.

**Answer**: Probably it would be more accurate to state that the African Easterly jet is weakened and pushed northward, since a strong westerly wind response is visible to the south and a weak easterly wind response exists to the north of the displayed African Easterly jet in the maximum lake and wetland experiment (Figure 5 b-c).

**Referee**: Would "ground albedo" be a better term than "background albedo"?

**Answer**: We will use the same term as Vambourg et al. (2011), because 'ground albedo' might have a different meaning than 'background albedo'.

**Referee**: L185 involve -> induce (?)

**Answer**: Okay.

**Referee**: L190. Can you explain the final sentence a little more? You don't have a figure showing heterogeneous rainfall in the paper, so this is a little complicated to grasp.

**Answer**: We agree that last sentence sounds too speculative. We therefore will remove it.

**Referee**: L195. You might also want to cite the compilation of Shanahan et al (2015).

**Answer**: We will add this citation to the manuscript

**Referee**: L231. I believe there is a missing word between "access water". Perhaps you could rephrase the sentence

**Answer**: Unfortunately, there was a typo in the originally submitted manuscript. What we had meant to write was "excess water". We trust that the meaning will be clear with the revision.

---

## Author Response (AR1)

**Letter to the editor cp-2021-129, "Simulated range of mid-Holocene precipitation changes from extended lakes and wetlands over North Africa" by Specht et al.**

Dear Qiuzhen Yin,

We have revised the manuscript according to our point-by-point reply to the referee's comments. For the sake of completeness, we add the referee's comments and our reply below. We considered all comments and suggestions by the referees. We also repeated all simulations and analysis described in the study due to a bug found in the boundary conditions. The bug affected the results only marginally, and our conclusions remain unchanged.

**Referee**: Thanks very much for assigning me to review this manuscript entitled *"Simulated range of mid-Holocene precipitation changes to extended lakes and wetlands over North Africa"*. The authors analyze the distinct responses of mid-Holocene precipitation to prescribe several surface boundaries conditions. Overall, this paper very interesting and the logical is clear. From my understanding, moderate revisions are needed before its publication on this journal.

**Answer**: Thank you for pointing out that the study is interesting and that the main story line is clear.

**RC1**:

Comments on this manuscript:

For the method part,

The authors should give more details of "moisture budget analysis and its thermodynamic and dynamic decompositions"used in this study, because not every reader is a good at dynamics.

**Answer**: We agree that a more detailed introduction of the moisture budget components would support the comprehensibility of the manuscript. Therefore, we will include the equation by Seager et al. (2010) into the methods part and explain these terms using simple examples, e.g. "the dynamic term prescribes changes in the moisture budget caused by a response in the mean circulation, e.g. changes in inland moisture transport by the monsoon westerlies."

**RC1**:

For the result part,

Please give one more Fig2b in the Fig2 which is related to"evaporation "responses, in order to support the authors statement "P4L108-110"
**Answer**: We will add an additional figure (map plot) that show the contribution of the local evaporation response and moisture budget response to the overall precipitation response.

**RC1**: Please use "local evaporation"instead of "direct evaporation"(P5L123)
**Answer**: Will be done.

**RC1**: Please take care the statement (P5L125). Indeed was neglected in Seager et al. 2010.

**Answer**: I am not sure what this comment refers to, but we will take care of explaining the moisture budget analysis and its meaning in the method part.

**RC1**: Please check the order of the Fig.3 and Fig.4, I am afraid the fig.3 should have appeared in some place prior to Fig.4.

**Answer***: We will changes the order of figure 3 and 4.

**RC1**: If possible, please add the extent of change in each experiment so that it will Convery a very clear picture for the reader.

**Answer**: We will show the contribution of the local evaporation response and moisture budget response to the overall precipitation response in a separate figure (as mentioned above).

**RC1**: Typo error, ITZC L149

**Answer**: We will correct this typo.

**RC1**: Please add the horizontal circulation changes in each subplot of Figure 4 to support your statements P6L133-134

**Answer**: We will add the horizontal wind response at 850 hPa to figure 4 d-f.

**RC2**: This paper is obviously a good fit for publication in climate of the past. It describes some novel simulations looking at the role of lakes and wetlands over North Africa. There is clearly sufficient work represented by simulations to warrant a publication, and I commend the authors for having done these runs. The inclusion and analysis of the water budget is also a nice feature.

**Answer**: Thank you for the positive feedback on the simulations and the moisture budget analysis.

**RC2**: Unfortunately I don't feel this manuscript is worthy of publication in its current state, because of the open questions around seasonality. A quarter of the manuscript's Discussion section is devoted to speculation about the rainfall changes in autumn using other published work. This is further emphasized in the conclusions. If I've understood the methodology section correctly, then such information could be computed directly by the simulations. A revised manuscript should include some analysis of this data for autumn – preventing there being any need to speculate.

**Answer**: We agree that our discussion was too speculative. Therefore, we will include additional analysis on the seasonality of the precipitation response from the prescribed lakes and wetlands to provide a more comprehensive picture. The additional plot will show a substantial precipitation increase over Northwest Africa around the end of the rainy season.

**RC2**: Other minor comments.

The title should be reworded to say "from extended lakes", to better convey the direction of influence.

**Answer**: We will reword the title as suggested ("to" → "from")

**RC2**: I suggest changing lines L42-L44 from

"Moreover, little research has been done regarding the role of vegetated wetlands during the mid-Holocene (Carrington et al., 2001). Present investigations on the effect of vegetated wetlands prescribed a small extent in the western Sahara and in the vicinity of mega-lake Chad (Carrington et al., 2001; Hoelzmann et al., 1998). Apart from mega-lake Chad,"…

To

"Moreover, little research has been done regarding the role of vegetated wetlands during the mid-Holocene. Vegetated wetlands have been prescribed in the vicinity of mega-lake Chad (Carrington et al., 2001; Hoelzmann et al., 1998), yet"…

**Answer**: We rephrase this test part as suggested.

**RC2**: L47 "in mid-Holocene" should be "of mid-Holocene"

**Answer**: We will rephrase this text part as suggested.

**RC2**: L60 Whilst I appreciate you citing both of my papers here, I am unclear why our work on ENSO is relevant. Instead, you might want to cite the doi number from the EGSF to give better credit to the MPI team who performed the mid-Holocene simulation.

**Answer**: Thank you for making this point. We have decided to take the SST and SIC from the PMIP simulation published in your paper. We will add citation of the DOI provided by the ESGF in the method part and data availability and we will acknowledge Roberta D'Agostino who did this simulation. Regarding vegetation, we choose to take the vegetation distribution, including the patterns of plant functional types, from a parallel 6k simulation, which is published and described in Dallmeyer et al. (2021).

**RC2**: L65 What is it necessary to the monthly SIC and SST anomalies from the MPI mid-Holocene simulation to your experimental design. I can see why you might do it, but you analyse nothing but the climatologies in this paper.

**Answer**: From our experience, we know that the simulated atmospheric mean state differs depending on whether only climatological SSTs and SICs were prescribed or climatological SSTs and SICs plus variability.

**RC2**: Fig 1. Add in the caption that the present-day lakes are also drawn. Can you make them more visible as well, please?

**Answer**: The Chad outline shown in figure 1 is drawn by default by the plotting tool NCL and does not show the present-day Chad lake prescribed in the control experiment. We will therefore add a third plot that shows the present-day lake distribution as prescribed in the experiments.

**RC2**: It is not clear how eq. 1 differs from that of Egerer et al. Please explain.

**Answer**: We will add the information that Egerer et al. 2018 uses the following linear approach: $MIN(NPP/NPP_{soil}, 1)$; whereas an inverse exponential approach is used in our study: $e^{\wedge}(-NPP/NPP_{soil})$. This was already stated in the manuscript, but obviously not explained very clearly.

**RC2**: L104: prescribed -> proposed

**Answer**: Okay.

**RC2**: L115: Please add "likely" before associated. You provide no evidence of this statement.

**Answer**: We agree on putting "likely" before statement L115

**RC2**: L116: It is unclear what evidence you are basing this claim on. Figure 2 shows only zonal mean values, and so cannot be used to support discussion of regional changes.

**Answer**: We will exclude this statement for the sake of clarity.

**RC2**: Figure three is barely mentioned and feels like a remnant of an earlier draft. It could be a useful figure.

**Answer**: One of our co-authors, whose daily business is not the AHP, requested this figure to improve clarity for the non-initiated readers.

**RC2**: L155. The "Sahara heat low" is an undefined term. I find myself thinking of low heats and cold places.

**Answer**: The Saharan heat low is a standard meteorological term (e.g. Claussen et al. 2017; Nicholson, 2009) .

**RC2**: Please label Fig 4 as mm/day.

**Answer**: Okay.

**RC2**: L165: figure 5 doesn't really show a northward shift of the African easterly jet. Rather it shows a death of the southern portion of it.

**Answer**: Probably it would be more accurate to state that the African Easterly jet is weakened and pushed northward, since a strong westerly wind response is visible to the south and a weak easterly wind response exists to the north of the displayed African Easterly jet in the maximum lake and wetland experiment (Figure 5 b-c).

**RC2**: Would "ground albedo" be a better term than "background albedo"?

**Answer**: We will use the same term as Vambourg et al. (2011), because 'ground albedo' might have a different meaning than 'background albedo'.

**RC2**: L185 involve -> induce (?)

**Answer**: Okay.

**RC2**: L190. Can you explain the final sentence a little more? You don't have a figure showing heterogeneous rainfall in the paper, so this is a little complicated to grasp.

**Answer**: We agree that last sentence sounds too speculative. We therefore will remove it.

**RC2**: L195. You might also want to cite the compilation of Shanahan et al (2015).

**Answer**: We will add this citation to the manuscript

**RC2**: L231. I believe there is a missing word between "access water". Perhaps you could rephrase the sentence

**Answer**: Unfortunately, there was a typo in the originally submitted manuscript. What we had meant to write was "excess water". We trust that the meaning will be clear with the revision.

---

## Author Response (AR2)

**Reply on Referee's technical comments on cp-2021-129, "Simulated range of mid-Holocene precipitation changes to extended lakes and wetlands over North Africa" Specht et al.**

We want to thank Chris Brierley for his valuable and precise comments on the revised manuscript version. In the following, we will answer to these comments point-by-point.

I accept that "Saharan heat low" is a legitimate meteorological term, but I still think that a short explanation or citation for it would be helpful on L49 – for readers who are e.g. palaeoceanographers.

**Answer**: We added a brief explanation to L49 with 2 citations from literature.

You need to reconsider your data availability statement. It currently only discusses the availability of the data inputs for the simulations – rather than the outputs. You must provide at least provide the data that is plotted in your figures.

**Answer**: We added a link to the section "data availability" that makes the relevant data available for the readers. This includes a description of the model version, the boundary conditions, the run scripts, and the output data used in our study. Unfortunately, this additionally link is not visible in the latexdiff version for some reason.

I have also spotted minor changes that might be worth implementing before typesetting:
- L9. I suggest that "the isohyets of" is removed. This technical term adds little to the sentence, but may put off potential readers.

**Answer**: We removed the term as suggested.

- Figure 2. Can you add something about the 200 mm/year reference line to the figure caption, please.

**Answer**: We made the requested addition to the figure caption.

- Figure 3. This is a lovely figure now, but the caption needs to editing. It currently leaves the identification of the various runs to panels e)-g). Can you also explain the black line in the hovmuller as that is important to your conclusions. Smooth -> smoothed.

**Answer**: We extended the description in the caption and we explained the meaning of the black line in plot a-d. We also changed "smooth" to "smoothed".

- L180. Lake Chad -> Lake Chad

**Answer**: We changed lake Chad to Lake Chad.